# The possibilities of the use of N-of-1 and do-it-yourself trials in nutritional research

**Tanja Krone**, **Ruud Boessen**, **Sabina Bijlsma***, **Robin van Stokkum**, **Nard D. S. Clabbers**, **Wilrike J. Pasman**

TNO, Zeist, The Netherlands

* sabina.bijlsma@tno.nl

## Abstract

### Background

N-of-1 designs gain popularity in nutritional research because of the improving technological possibilities, practical applicability and promise of increased accuracy and sensitivity, especially in the field of personalized nutrition. This move asks for a search of applicable statistical methods.

### Objective

To demonstrate the differences of three popular statistical methods in analyzing treatment effects of data obtained in N-of-1 designs.

### Method

We compare Individual-participant data meta-analysis, frequentist and Bayesian linear mixed effect models using a simulation experiment. Furthermore, we demonstrate the merits of the Bayesian model including prior information by analyzing data of an empirical study on weight loss.

### Results

The linear mixed effect models are to be preferred over the meta-analysis method, since the individual effects are estimated more accurately as evidenced by the lower errors, especially with lower sample sizes. Differences between Bayesian and frequentist mixed models were found to be small, indicating that they will lead to the same results without including an informative prior.

### Conclusion

For empirical data, the Bayesian mixed model allows the inclusion of prior knowledge and gives potential for population based and personalized inference.

**Data Availability Statement:** All relevant data are within the paper and its Supporting Information files.

**Funding:** The author(s) received no specific funding for this work.

**Competing interests:** The authors have declared that no competing interests exist.

## Introduction

The current trend towards personalized medical treatments and life style advice is also evident in the field of food and nutrition [1,2,3,4]. Research on the effectiveness of specific foods or ingredients to tackle an individual health problem is, however, not straightforward. For example, the difference in metabolism may influence the effect of caffeine, or the effect of dietary restrictions on well-being and weight. As such, the field of nutritional studies is evolving towards individualized analyses. This hold implications for the methods used in both data gathering and data analysis.

A common approach in nutritional research up until now is the use of randomized controlled trials (RCT). The aim of these trials often is to investigate treatment effects on group or population basis. Two types of frequently applied RCT are crossover trials and parallel designs [5]. In crossover trials, subjects are assigned to a sequence of at least two conditions of which one generally consists of a control (e.g.: no treatment). Parallel designs, in contrast to crossover designs, have distinct control and treatment groups. Subjects participating in the trial are randomly allocated in either the treatment or control group. This allows comparison of the treatment effect against the absence of the treatment (the control), assuming a similarity between the two groups due to random allocation of the subjects.

Traditionally, RCT's are used most often in nutritional research [5,6]. However, several drawbacks of these studies urge us to look to further methods. First, RCT findings can be hard to generalize beyond the clinical setting into real life applications, referred to as a lack of so-called 'ecological validity' [7]. Second, RCT is focused on group comparison for the estimation of treatment effects on the population level [8]. These kind of trials therefore are not designed to obtain individual deviations from the population required for personalized nutritional advice.

Driven by the need for a more realistic research setting, new technologies and community movements, methods like Quantified Self (quantifiedself.com), Do-it-Yourself (DIY) trials and self-experiments are introduced in the nutritional research field [9]. In these kind of studies participants do most, if not all, of their measurements themselves using monitoring applications or devices (e.g. smart phones), called self-measurements. Often, these measurements take place in their everyday environment. If these measurements are done in a relatively high intensity in time, we talk about ecological momentary assessment (EMA, also known as experience sampling method, 7, 10). Self-measurements may be used to increase ecological validity, as they support health claims based on real life situations instead of clinical settings [10,11].

As self-measurements allow for high intensity information gathering, it is possible to create so-called N-of-1 trials. N-of-1 trials, also known as single subject or alternating treatment designs, allow for high-intensity data gathering on a single individual, tailoring to the individual changes over time. They have a long history in behavioral science but are increasingly applied in epidemiology and medical research as well [12,13,14,15].

The purpose of N-of-1 trials using self-measurements in nutritional science is generally one of two. Either to investigate the effect of some nutritional product on a certain outcome measure over time, or to be able to define and follow-up on personalized advice (e.g.: 16). The purpose of the first application is comparable with the classical approach in nutritional research, but potentially with lower costs and with a lower burden for the participants. The second application makes it possible to gap the bridge between the clinical setting and a more realistic situation, which fits well within the growing interest for personalized health targets and personalized (nutritional) advice.

A typical way of employing N-of-1 trials is to compare two or more experimental conditions (e.g. (therapeutic) treatments, foods, behavioral interventions) on an individual subject

[3,16,17]. The design often involves multiple consecutive periods as in cross-over trials in which the studied conditions are presented to the subject. The order of presentation may be controlled by the researcher, to overcome potential confounding by order effects, underlying time-trends, and/or carry-over effects, in which effects of one experimental condition are carried over to the next experimental condition [11,18]. Where not possible, this may be taken into account with the analyses later on. When analyzing the data resulting from these N-of-1 trials, a more individualized approach to data analysis is also needed.

The analyses of N-of-1 trials, especially those done using EMA studies, are dependent on specialized statistical methods. Several statistical methods have been proposed in recent years, of which Individual-participant data meta-analysis [IPD, 19, 20, 21] and linear mixed effect models (also known as hierarchical or multilevel) models are often used. The most important difference is that IPD takes a two step approach in calculating the expected effect on populations levels. IPD creates a linear model for each individual separately, after which it combines the results it find. In linear mixed effect model, this combining is done right away: the data is analyzed together. Both models thus allow for individual deviations, but create a model meant for the whole population. For the linear mixed effect modelstwo popular approaches are frequentist linear mixed modelling (F-LME), using estimation techniques such as maximum likelihood estimation [22, 23, 24, 25] and Bayesian linear mixed modelling (B-LME), using estimation techniques such as Markov chain Monte Carlo (MCMC) [26, 27, 28].

Several studies have compared IPD and linear mixed effect models for N-of-1 data. For example, Zucker et al. applied IPD and LME-F to an empirical dataset of 46 N-of-1 trials with 2 to 6 periods of treatment with either only amitriptyline or amitriptyline combined with fluoxetine as a cure for fibromyalgia syndrome [28]. Zucker varies elements of the models to find which model comes closest to the observed data. However, since the true population treatment effects were unknown, an objective comparison standard is missing. Consequently, it was not possible to compare these methods in terms of performance under different circumstances.

More theoretical approaches have been taken to compare these ways of analyzing N-of-1 data, by simulating a fully crossed design with varying time series and number of individuals. These results indicated that F-LME and B-LME performed similarly and generally outperformed IPD methods [29, 30, 31]. However, these studies did not include different treatment effects but only an time-dependent effect known as autoregression or autocorrelation. Another simulation study in the context of investigating IPD showed mixed results, showing a underestimation of the differences between persons [32]when using IPD. However, this study did not include mixed models.

In this paper we will show the added effect of using an inclusive method on all data together, where possible and the influence of priors on the results. As opposed to early studies, we will use different scenarios to show the different ways in which the models react. In the next section we will explain the analysis methods and the differences between them. We will continue with a simulation study comparing the methods. Following this, we will demonstrate the use of the B-LME using different priors on an empirical dataset.

## The three statistical models

### The individual participant data meta analysis

The first of the three statistical models we will elaborate is the individual participant data meta analysis (IPD). In general, meta-analyses combine the results from separate but similar studies to allow a more precise and reliable estimation of a common effect of interest [33]. In combining N-of-1 trials holding several measurements over time per individual, which can be seen as individual but similar studies, this method aggregates the individual effects. First, a model is

created for each individual, for example, a linear model. After this, the results from these models are combined to create a population effect. The estimation of the population effect from the individual effects can be done in several way, but the simplest approach is to take the mean of the individual effects. This approach can be used for any of the effects modelled, and also for the error variance. Combining these models can be done by applying weights to the studies, dependent on for example the study size [20].

## Linear mixed effect models

When we aim to analyze all individuals in one model, while retaining the differences between the individuals, the linear mixed model is often used, also known as hierarchical or multilevel models. In our case they consist of two levels. The lowest level consists of the individual measurements *nested* within the individual. The second level consists of the respondents *nested* within a (treatment or control) group, as depicted in Fig 1. It is true that this could also be seen the other way around in crossover trials; measurements nested within treatments which are nested within individuals. However, this would imply we are more interested in the differences between persons then in the differences between treatments. Mixed effect models are well suited for modeling the differences between persons and the longitudinal (sequential) aspects of the data resulting from the setup of N-of-1 trials [34,35].

When using a linear effects model, it is assumed that the properties of the nested group, are related to the group properties, e.g. the treatment effect of all individuals combined show a normal distribution around a population mean. Thus, by including several individuals nested in a group in a single mixed model, both an overall group effect and an individual effect can be estimated. The variability (i.e. standard deviation) of this distribution informs us of the variation of effects between individuals [34,35]. If the assumption of nested properties holds, the model has more power than the individual meta-analysis model.

The estimation method is key in the difference between Frequentist linear mixed effect models (F-LME) and Bayesian linear effect models (B-LME); the first generally uses maximum likelihood estimation and the second Monte Carlo Markov Chain estimation. A more detailed explanation of estimation methods is beyond the scope of this paper. For more information, please see Hox and Gelman [36,37].

When there is information on the expectations of parameter values before the analyses is conducted, this may be included in the prior of the model. This is what distinguishes the Bayesian mixed model from its frequentist counterpart conceptually. Incorporation of prior information brings the benefit of additional power and robustness in estimating coefficients

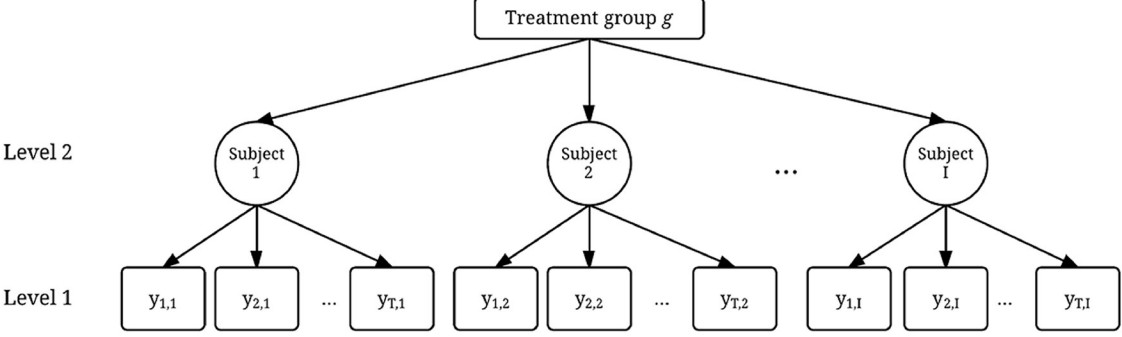

**Fig 1. A generalized form of a multilevel structure for one group.** The measurements are in level one for time point t = 1, 2,. . ., T for subject y = 1,2,. . .,N. The subjects are level 2 and combined in one treatment group.

[34], especially for small datasets. An example is including prior information on the treatment effect. Consider an intervention study with body weight as its primary outcome, where a comparable study had previously shown that subjects with a baseline body weight above 100kg all benefit from a low-calorie diet while only half of the population with a baseline body weight below 100kg does. In this case a prior on the treatment effect for subjects having a body weight above 100kg could be formed with emphasis on the assumption that they all lose weight. In case no prior information is included (known as a non-informative prior) the analysis gives the same numerical results as the frequentist mixed model.

## Methods of simulation study

### A linear mixed model for N-of-1 trials

A model that may analyze N-of-1 data to such an extent that it is useful in the situations described in the introduction, must meet certain criteria. These are based on some assumptions of the population to be studied. First of all, individuals have different starting points, which means different intercepts. Second, not everyone reacts the same to the treatment. There are two approaches of including this: we assume it is based on a known variable, such as starting weight, or we assume we do not know exactly why this is. As we generally do not know this in studies, we'll assume there is a unknown effect, creating a individual differences within the population with a normal distribution. Third and final, the effect of the treatment must be independent of time, e.g.: the effect size in week 3 must be equal to the effect size in week 10. This is not realistic, we understand this. However, it simplifies both the simulation method and the analyses done later in this paper. This way, the order of the conditions is irrelevant, and the effect of one condition is expected to be washed out before the start of the next condition.

The above points bring us to a linear mixed effect model: we assume some unknown but persistent variations between individuals in the model. The statistical formulation of this linear mixed effect model for a situation with one treatment and one control condition, considering a continuous outcome variable $Y_{ij}$ for subject $i = 1,., N$ at time point $j = 1,., J_i$, is given by:

$$Y_{ij=} \mu_0 + \mu_i + X_{ij}\beta_i + \varepsilon_{ij}, \tag{1}$$

and

$$\varepsilon_{ij} \sim N(0, \sigma^2), \tag{2}$$

$$\mu_i \sim N(0, \tau^2), \tag{3}$$

$$\beta_i \sim N(\beta_0, \omega^2). \tag{4}$$

where $\mu_0$ is the general intercept representing the baseline population level, e.g. the overall average body weight, and $\mu_i$ the person-specific deviation from the general intercept, e.g. how much heavier (or lighter) a person is compared to the overall population average weight, which follows a normal distribution with mean zero and variance $\omega^2$ (Eq 3). The allocation of treatment and control at each time point is described by an indicator (or dummy) variable $X_{ij}$ (0 for control, and 1 for treatment), and $\beta_i$ is the corresponding person-specific treatment effect relative to the control condition (e.g. the increase/decrease in body weight for a specific person as a result of the treatment), distributed with mean $\beta_0$, the average treatment effect, and variance $\omega^2$ (Eq 4). Finally, we assume that $\varepsilon_{ij}$ is independent of random effects $\mu_i$ and $\beta_i$. Note

that this notation allows for extension to more than a single treatment group by including indicator variables.

The model specified by Eqs 1–4 is a general mixed effect model which allows for heterogeneity in the baseline (the intercept) and treatment effects. Usually, as in the model specified above, this heterogeneity is modeled as the deviation from the average or fixed effect. This is shown in Eqs 3 & 4, where the variance components $\tau$ and $\omega$ specify the amount of variability around the average treatment effect $\beta_0$.

For both the linear mixed effect models and the IPD, we will use model Eqs 1–4. However, the estimation is a two step procedure for the IPD, where we estimate the individual models and then aggregate them by taking a weighted mean for each parameter. As such, the standard deviation of the error $\sigma^2$ may change between individuals, and the variance of $\mu_i$ and $\beta_i$ are not explicitly estimated. In this simulation, the weight for each individual will be one, as all participant will be included for the same amount of measurements.

## Testing hypotheses

N-of-1 data may be used to test several kinds of hypotheses. The homogeneity of an effect within a population can be assessed, e.g., is the weight loss for every individual similar, or we may test hypotheses on the direction e.g., do all individuals benefit from an increase intake of iron. With N-of-1 trials, a treatment effect can be estimated for a person only if he or she receives both treatment and control conditions. A control condition may also be a "baseline": measurements taken before treatment starts.

As indicated before, in crossover-trials we need to assume that the effects of treatments washes out before the next treatment is administered, and that no systematic time-trend is present (e.g. because of ageing of the person, adaptation to the treatment, or selection of the most-favored option by the person). Under these assumptions, one may infer whether *this* treatment has an effect for *this* person.

To illustrate the applicability of the three methods for analyzing N-of-1 trials, we present two examples. In the first, simulated example, we compare the performance of the meta-analysis method and the mixed models against each other. In a second example, based on a real dataset, the effect of including prior information in the analysis will be demonstrated.

## Design of the simulation study

A simulation study can be seen as an experiment where the focus is not on investigating a dataset using a fitting model, but on investigating a model using a fitting (often generated) dataset. By generating a population (and thus the subsequent sample), one knows all its properties which allows the researcher to investigate how well the model performs [38]. This simulation study aims to compare a) a two step approach with a one-step approach, and b) Bayesian with Frequentist estimation methods. We use different scenarios with different data properties, to compare the models, to show the merits of the models compared to each other, allowing a model choice based on data specification.

The nine scenarios were defined by varying two parameters; (i) the number of N-of-1 trials, which is the number of subjects, ($n$) set at 20, 30 or 40, and (ii) the intended number of measurements per subject ($t$), set at 10, 20, or 30. Other parameters we set (but did not vary) included the mean of the control condition ($\mu_0$) at 0 and variance $\tau^2$ of $\mu_0$ at 1, and the average treatment effect $\beta_0$ at -2 and the accompanying variance $\omega^2$ at 0.5. The error variance $\sigma^2$ was also set at 1. These scenarios help to determine whether the comparative performance of the models was affected by the simulation parameters we varying between scenarios. To conclude,

the steps making up the simulation experiment were repeated 1000 times for each of the nine scenarios to acquire a stable and reliable basis for comparison.

To create a dataset, we held on to the model as specified in Eqs 1–4. In every run, a data set was constructed through the following steps: first, each individual obtained a subject specific intercept, $\mu_i$, from a random normal distribution as defined by [3]. For each individual $n$, a series of measurements was created of length $t$, with mean $\mu_i$, and standard deviation $\sigma$, following Eqs 1 to 3, only leaving the treatment effect out. Next, the subject-specific treatment effect $\beta_i$, or the average different between treatment $i$ and the control condition, was drawn from the population effect distribution defined by $\beta_0$ and $\omega^2$, as defined in [4]. For measurement on treatment occasions, $\beta_i$ was added to the scores. Every subject had an equal number of measurements on treatment and control, but the order would (randomly) differ between subjects. The ratio of the error with the random intercept was 1:1 ($\sigma$: $\omega$), and with the random treatment effect 1:0.5 ($\sigma$: $\tau$).

## Results of the simulation experiment

### Used measures

The performance of a statistical model can be split in two important and distinct features: *bias* and *precision*. When an estimator is biased, it will give estimations that deviate in a certain direction from the true scores on the population level, i.e., it generally estimates that parameters higher or lower than they truly are. Lower bias is thus desirable. An estimator is deemed precise when its estimations show little variability, thus the estimations are closely centered around the true scores. Bias and precision can be combined as *accuracy*: when an estimator is said to have little bias and high precision, it produces accurate estimations.

Accuracy in this study will be quantified as the root-mean-square error (RMSE), a widely used measure of statistical performance [39,40]. The RMSE measures the absolute difference between an estimated parameter and its true value. A general formulation for the RMSE is:

$$RMSE = \sqrt{\frac{1}{q}\sum_{k=1}^{q}(\hat{\beta}_i - \beta_i)^2},$$ (5)

considering a set of RSME values for replication $k = 1,.., q$ where $q$ is the number of simulation repetitions, $\hat{\beta}_i$ is the estimated parameter of the model and $\beta_i$ is the true value in this study. RMSE is calculated by taking the root of the sum of squared differences of an estimate and its true value counterpart. By doing so, if effectively combines both precision and bias in a single measure. A smaller RMSE indicates higher accuracy. The RMSE was stored for all replications per scenario and formed the basis for model comparison.

### Results

Fig 2 presents the distribution of RMSE values for $\beta_i$ over all 1000 replications. It can be seen that the mixed models have a consistently lower RMSE than the IPD, indicating better accuracy. This is because the mixed models take into account the whole distribution, in stead of calculating the mean for each parameter. However, the difference between the mixed models and IPD decreases when the number of measurements per subject increases (going down over de rows in Fig 2). This is due to the fact that with an increasing number of measurements per subject, each individual obtains more power, and the distribution per individual draws closer to the assumed normal distribution. The difference between F-LME and B-LME are negligible. This indicates that Bayesian mixed models without informative prior perform approximately equal to their frequentist counterparts.

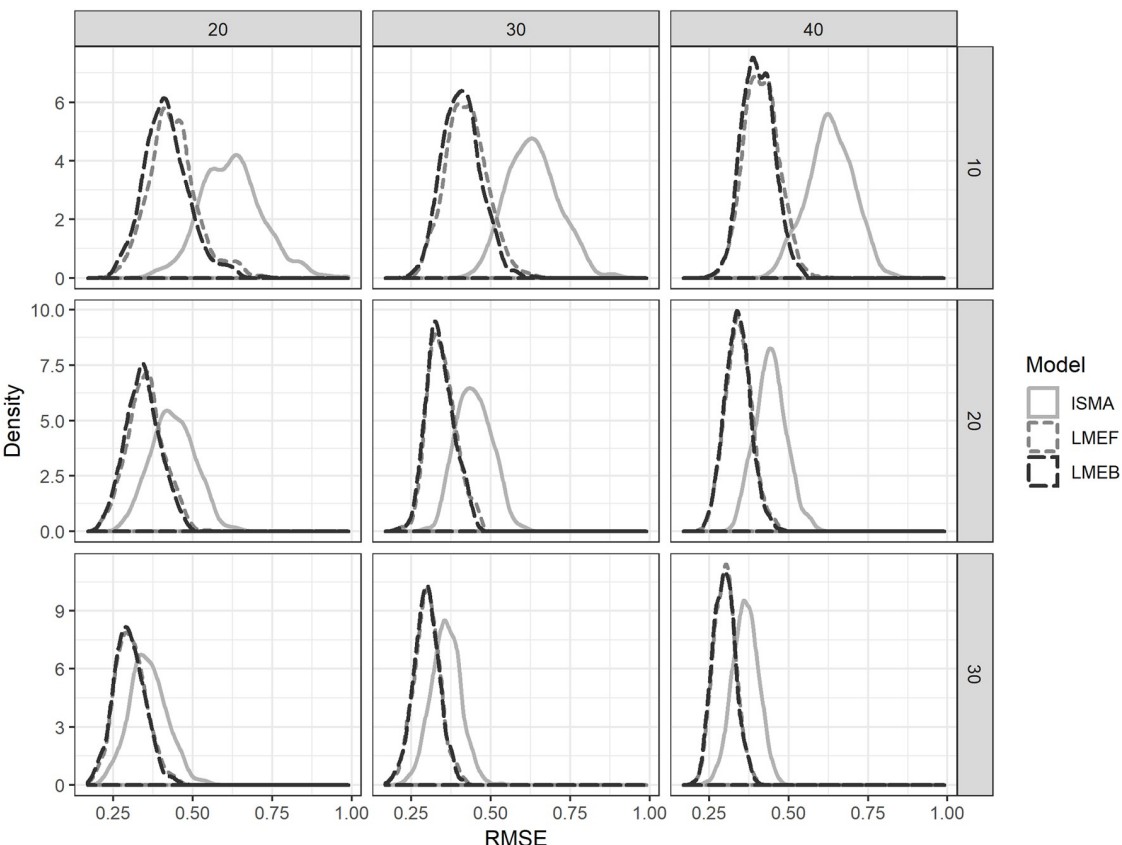

**Fig 2. Empirical distribution of RMSE values for $\beta_i$, the person-specific treatment effect relative to the control condition, over 1,000 replicates per condition for the individual-subject meta-analysis (ISMA) and the frequentist mixed models (F-LME) and Bayesian mixed models (B-LME) by number of subjects (columns; 20, 30 or 40 subjects) and number of measurements per subject (rows; 10, 20 or 30 measurements per condition).**

Fig 3 shows that the IPD variance estimation tends to be positively biased, meaning it over-estimates the variability in treatment effect $\beta_0$ across individuals, while the mixed models came closer to the true, simulated, value of 0.5. The IPD shows a smaller range of the estimated variance of the treatment, which can be seen from the lower variability in the estimation of $\tau^2$. In line with the previous results, LME-F and LME-B are approximately similar in their results. In practice this means that the mixed models are to be preferred over IPD, since the standard deviation in individual effects are estimated with lower bias, especially with lower sample sizes. A small bias is preferred over low variability, as a smaller bias will give you results closer to the underlying true value, while low variability with high bias will keep giving the wrong answers when replicated.

## Bayesian analysis of empirical data

To apply the discussed Bayesian method, we used a previously gathered dataset and compared the results for two different prior distributions.

### Methods

The Bayesian mixed model was applied to a series of 12 individual self-measurement trials. The aim of this example is twofold: 1) to demonstrate how prior knowledge may be included

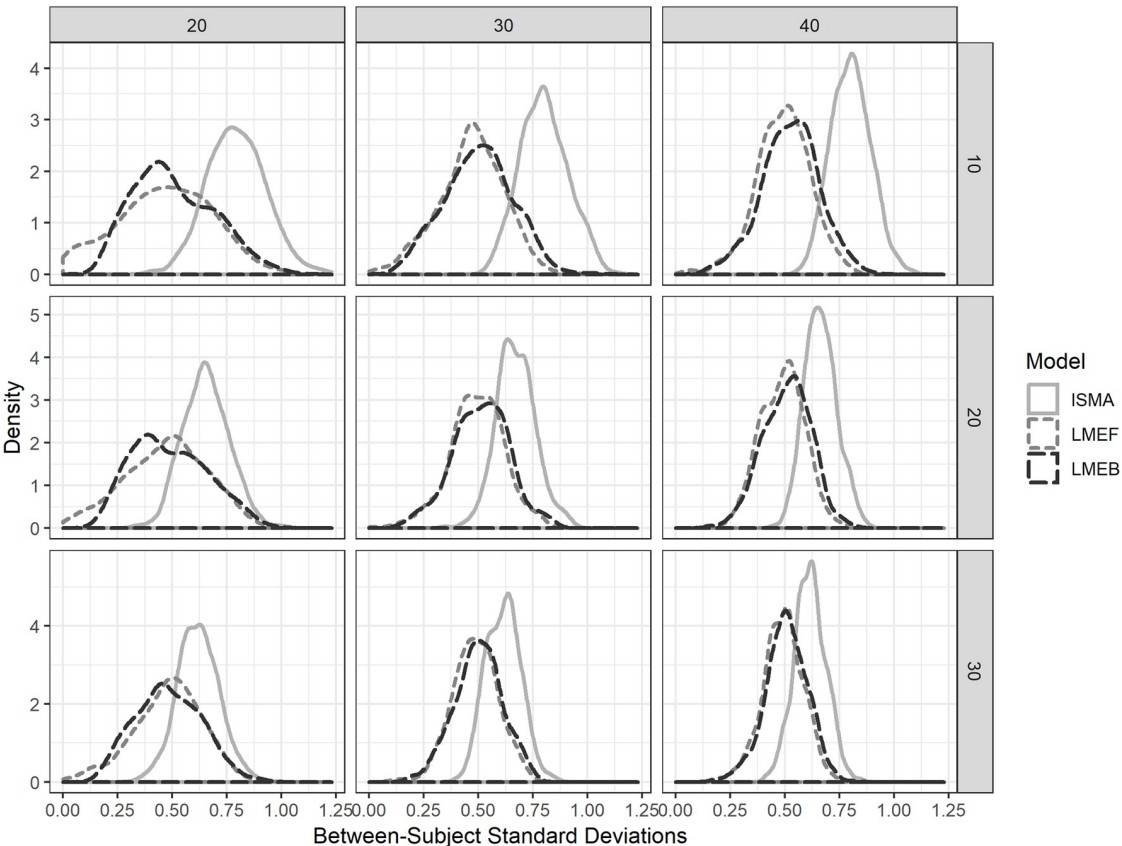

**Fig 3. Empirical distributions of the 1,000 replications of the estimated between-subject standard deviations of the treatment effects for the individual-subject meta-analysis (ISMA) the frequentist mixed model (F-LME) and the Bayesian mixed model (B-LME) by number of subjects (columns; 20, 30 or 40 subjects) and number of measurements per subject (rows; 10, 20, or 30 measurements per condition).** The average of the simulated true value of the standard deviation is 0.5.

and 2) to illustrate the possibilities for analyses of N-of-1 trials in more detail. The subjects included in thesetrials used wearable devices, scales and a food intake diary to monitor physical activity, body weight and macro nutrient intake during a 9-week period. They were encouraged to stick to their usual lifestyle during weeks 1–3, decrease the intake of high-caloric snacks during weeks 4–6 and increase physical activity during weeks 7–9. Subjects were males or females in healthy physical condition. The outcome of interest was the daily body weight measurement as collected by the subjects themselves. The possibility of including prior information in the analysis is demonstrated in two scenarios. The first scenario, the reference scenario, uses a prior distribution for the treatment effects variable that does not influence the results, known as a non-informative prior. The second scenario assumes that the two interventions will, on average, result in weight loss. It should be noted that this example merely serves to illustrate the model. There is much more to say about the study and its findings, but that is outside the scope of this paper.

Fig 4 shows the course of each individual self-measurement trial in the empirical example data set. As can be seen, useful, complete data was available of four males and eight females. One male and one female swapped the order of conditions two (i.e. reduce high-caloric snack intake) and three (i.e. increase physical activity). In addition nine subjects, two males and seven females, had at least one missing observation of which one female had as many as 41 missing observations out of the 63 possible measurements.

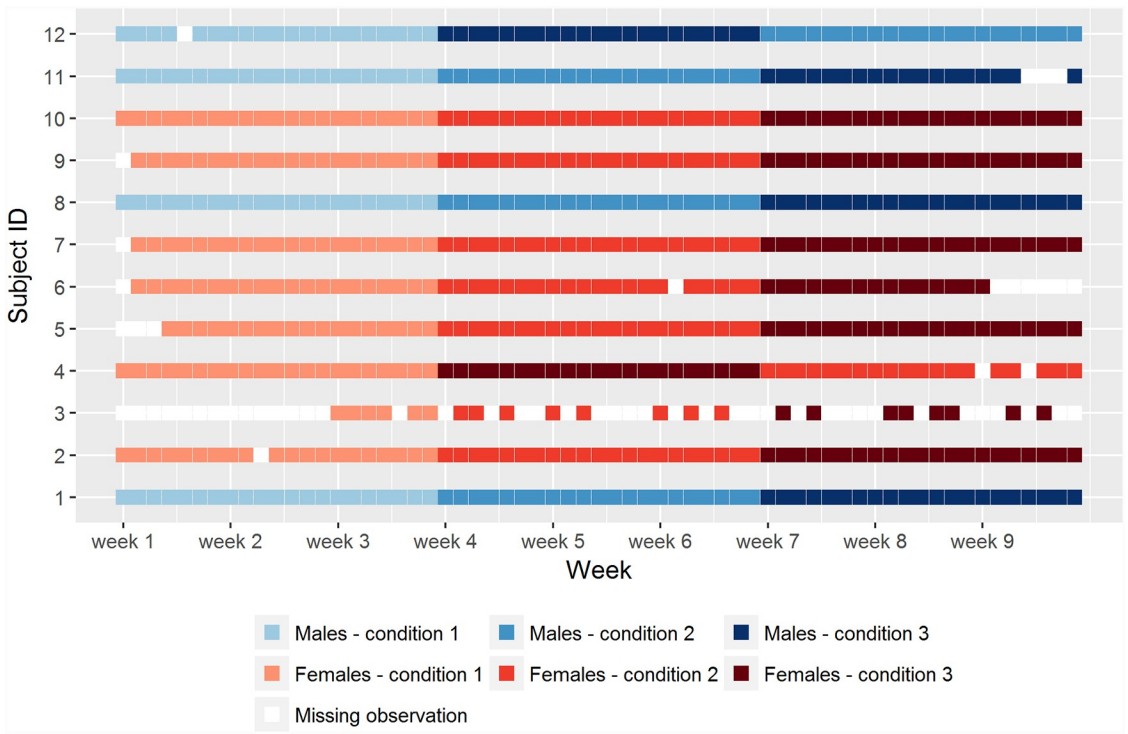

**Fig 4. The course of each individual self-measurement trial in the empirical example data set, the conditions are: Condition 1 = control period; condition 2 = less food intake; condition 3 = increased physical activity.**

Fig 5 presents the distributions of observed body weight measurements per week and per subject. These figures show the degree of variability between subjects and within and over weeks. One notable observation is the occasional occurrence of substantial variability in body weight within a given week, e.g. the first-week measurements of subject 1 (ranging from 68.7 to 71.7 kg.) or the third-week and fifth measurements of subject 12 (73.0–79.0 kg.). Another observation is the gradual decline in body weight over time as seen for certain subjects (most notably 6, 8 and 9, but to a lesser extent also 2, 4, 7 and 10) and an upward trend for individual 3. It is implied that the effect on body weight compared to condition is time dependent, i.e. the weight loss after condition 2 stays, and new habits may only be continued into condition 3. For now, we will not take this into account. In a real empirical study, the analysis would have been adjusted for this.

## Results

We present the results for the Bayesian mixed model for the weakly informative prior to illustrate the potential for population based and personalized inference from this dataset. The model parameters with their estimated range for these data are presented in Fig 6. The model shows an intercept combined with an effect for periods two and three separately for every subject. Period one is considered the baseline. It can be seen that the estimated efficacy of the interventions differed greatly over subjects. Several subjects showed a marked and significant estimated reduction in body weight for periods two and three. On the other hand, individual 5 is relatively stable over the three periods and individuals 1 and 3 are modestly increasing in body weight over the three periods. Exploring the individual estimates shows interesting differences amongst the individuals. Such differences in trends and effect sizes are not

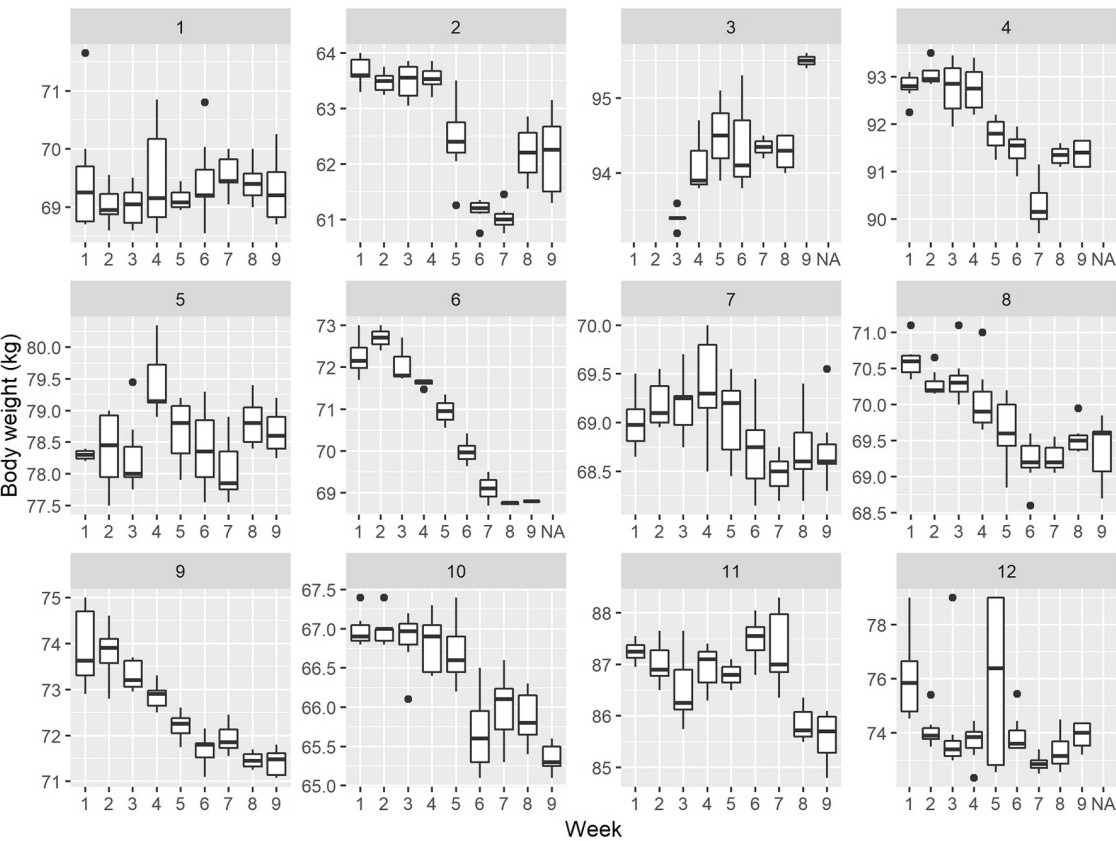

**Fig 5. The distributions of observed body weight measurements per week and per subject.**

immediately apparent from the population based estimates for the two conditions, which are shown in Fig 7. The example thus shows that it is possible to obtain population effect estimates, but also estimates of individual deviations from that general effect in the population.

## Demonstrating the inclusion of prior knowledge

To demonstrate the inclusion of prior knowledge in this example, the assumption was made that there would be a negative treatment effect for the population on body weight for the two conditions relative to the reference condition. In other words, it is expected that the average subject loses weight during condition 2 and 3. We compared two prior distributions for the population effect sizes of the two conditions: the reference scenario of a weak prior that expects an average increase of -1 kg and a standard deviation of 5 kg, which is a relatively flat and weak prior of which we expect no numerical influence on the results and which still allows both positive and negative effects for the population. The second prior distribution, the negative prior, entertains the same normal distribution, but truncated with an upper limit of 0 to exclude a population effect pertaining to weight increase, in accordance with our assumption. The priors for the two treatment conditions (i.e., increase of physical activity and reduced high-calorie snack intake) were thus the same, expressing that we do not expect an a priori difference between them.

The results for the population estimates can be seen in the boxplots shown in Fig 8. With the weak prior, positive weight increases are still plausible. However, the negative prior

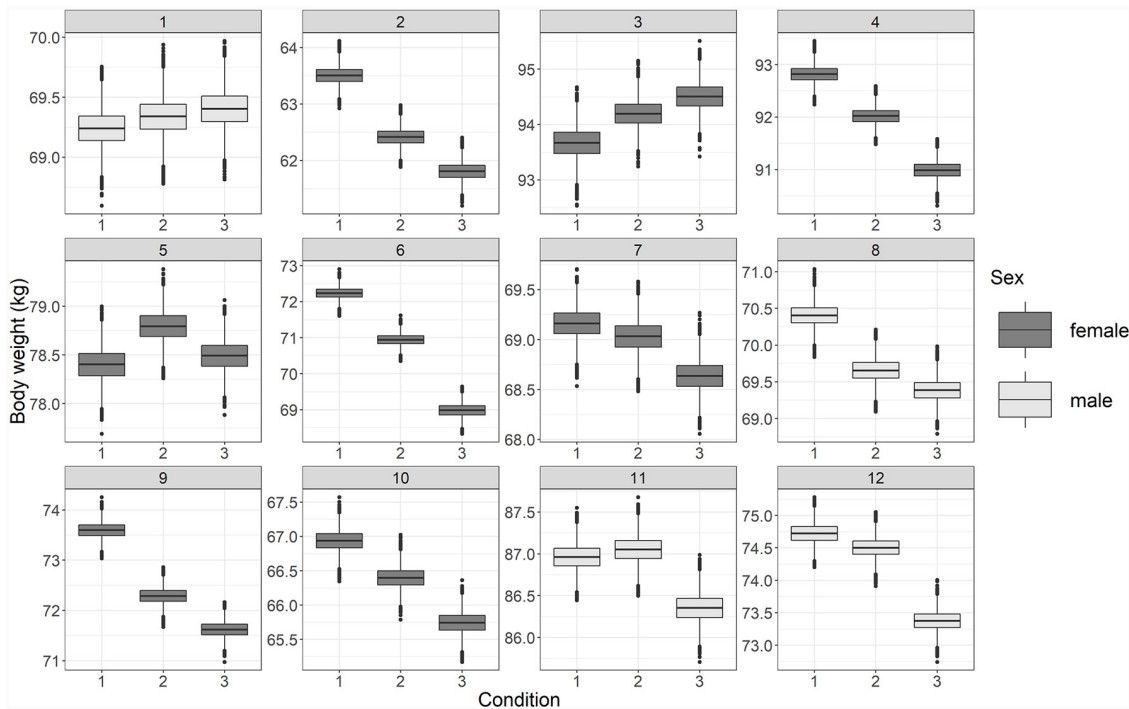

**Fig 6. The estimated body weight over the three periods for each subject.**

strengthens the results towards a negative weight increase for the two treatment conditions. Observe that the negative prior is still rather weak-informative, such that the expected treatment effects hardly differ. The results for the individual estimates (Fig 9) are hardly affected by the negative prior, which restricts the population effect to be 0 or negative, but not the individual effects.

## Discussion N-of-1 trial

This paper set out to demonstrate the ability of three popular statistical models to analyze N-of-1 trials, using both a simulated and empirical example. The simulation experiment showed that mixed models in general perform better than the individual meta-analysis method both on the individual and the population level, especially when the number of measurements per subject becomes smaller. This is in line with previous research, further supporting that mixed effect models have more power when the number of subjects is small [34,35]. Concluding, the simulation study showed that Bayesian and frequentist approaches perform similar when no informative prior is included, which is also in line with previous research [34,35,41].

The merits of the Bayesian linear mixed model in a typical research setting were shown using an empirical dataset. The analysis of the empirical example showed how individuals may differ in their expected treatment effects, while an average treatment effect for the two conditions suggested a population effect depicting weight loss for both. This is no novel discovery in nutritional research [e.g.: 42, 43], but it was shown that N-of-1 trials have the advantage of uncovering heterogenic responses on a treatment within a group of people. This can be helpful in figuring out the best personal treatment, in other words: tailoring the nutritional advice to the individual, as done for example by Madhok & Fahey [44]. Furthermore, the Bayesian approach to mixed models allows the inclusion of prior knowledge in a model, which may

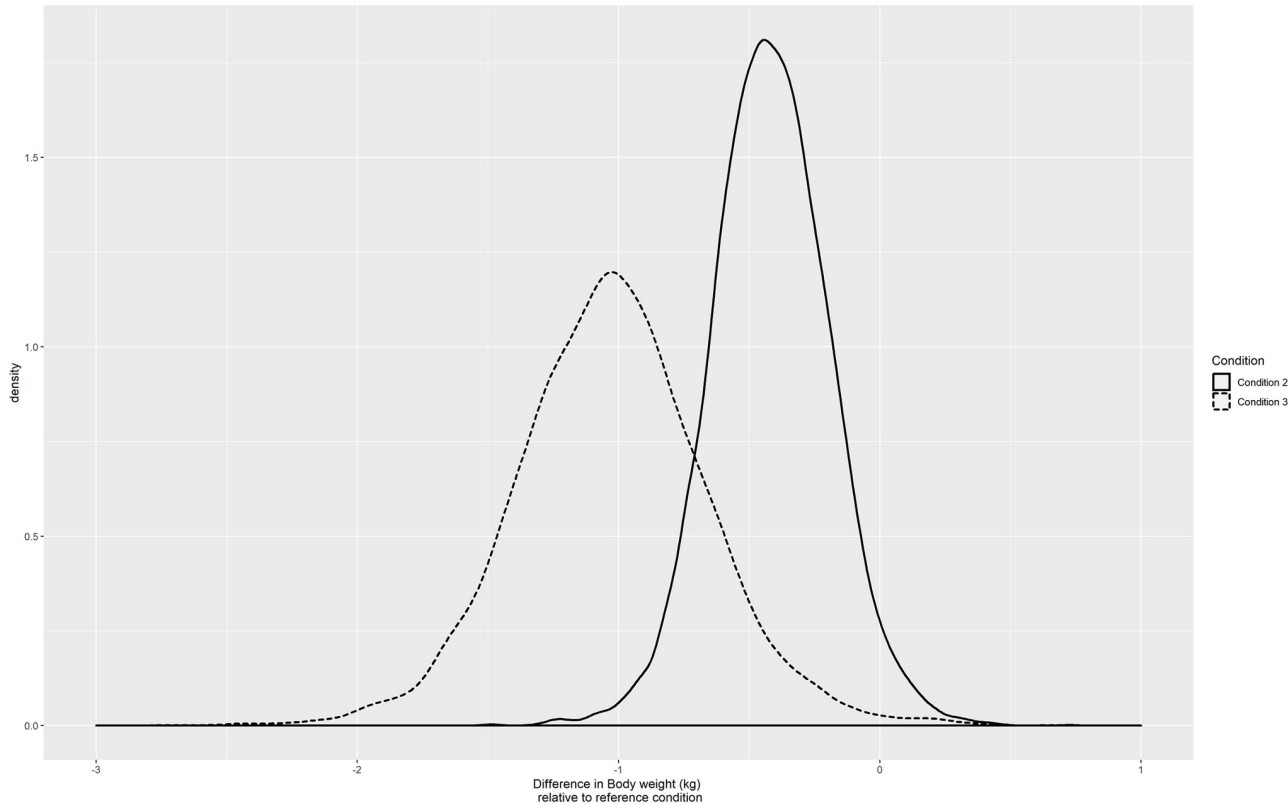

**Fig 7. The population estimates of the effects of periods 2 and 3 relative to the reference condition.**

enhance the power of the analysis due to the a priori restrictions on the parameter space or prior knowledge on (some of) the parameters. This can be especially helpful in exploratory studies with only a few participants to test a treatment, given some credible prior information is available.

The two studies had some limitations. The results of the simulation study can be seen as an indication of the performance of the models at hand. However, they are overly simplistic. The situation is stylized, meaning that the data adheres to the overall assumptions of the models. Real life applications rarely live up to the model assumptions made, and therefore some caution is warranted when generalizing the simulation results. Based on the results of this study, it is not clear how the three models perform when their assumptions are violated. Based on the literature, accuracy will be lost in both effect and variance estimations [41]. Finally, the simulation design was relatively simple, and it is thus difficult to assess how the models would behave in more complex, realistic situation, including for example trends over time and covariates in the model.

The empirical study had two important limitations. First, the intervention effect happened over time, i.e., the weight loss was not instantaneous and stable during the conditions, but happened gradually over time, violating the assumption of the absence of a systematic time trend. Second, the conditions of the study are not independent of each other. Weight is a relatively stable property and cannot be seen independent of the time leading up to the treatment. In other words, the order in which subjects received their treatments influences their response. These two violations underscore that real applications of statistical methods rarely meet their respective assumptions.

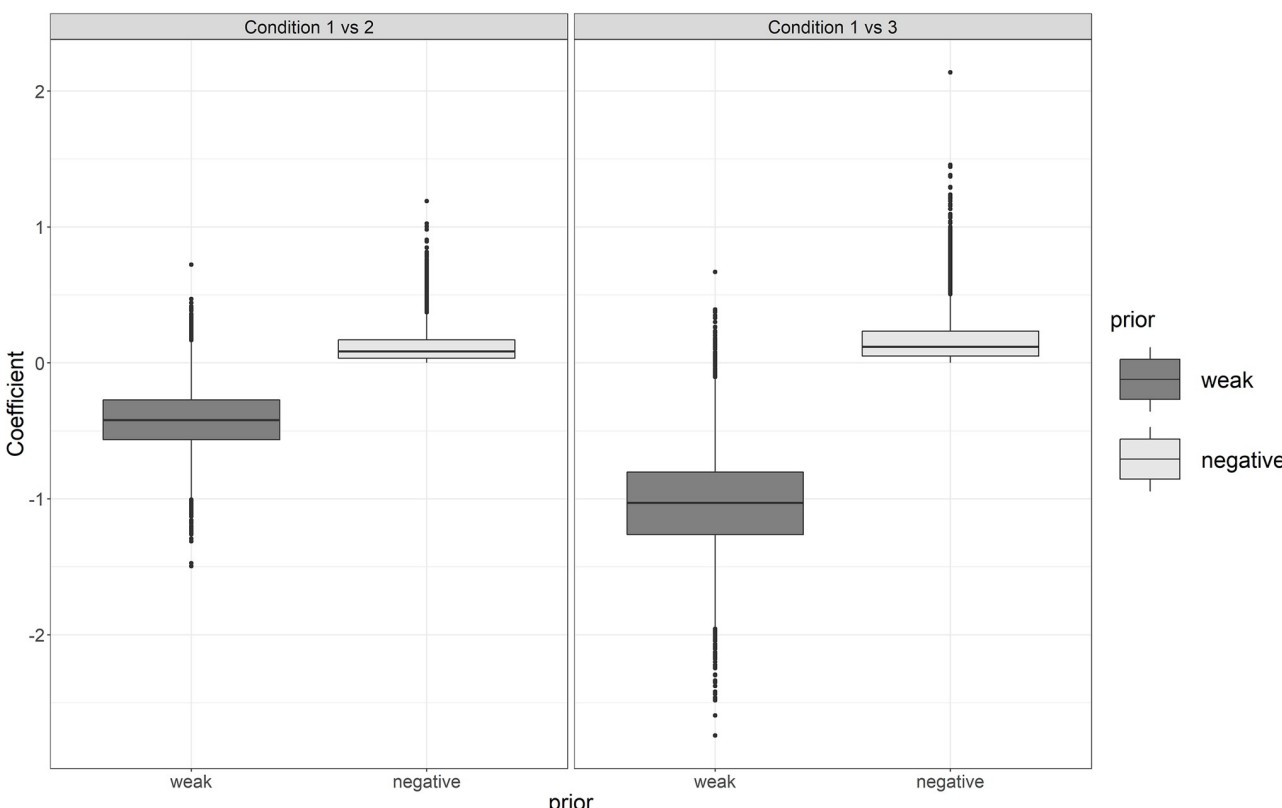

**Fig 8. Population estimates for the weight loss, using a weak and a negative prior.** Left: mean effect treatment 2 versus 1, right, mean effect treatment 3 versus 1.

A drawback of individual measurements based models, is that the estimation becomes problematic with few subjects, especially when the number of parameters in the model is substantial. An often used rule of thumb is that the critical point lies at a minimum of 10 individuals [45], preferably more than 25 to prevent underestimated standard errors and adequate confidence intervals [41]. With regard to timepoints, one must assess the number of time points per condition and not in total. The number of timepoints and the number of individuals cannot be seen separate: more timepoints means less individuals are needed, and vice versa. A cautionary advice for simple models, may be that at least one of the two must be 25 or higher, where the other must be 10 or higher for a stable, reasonably accurate model [30].

This paper further introduced the N-of-1 trial paradigm in the field of nutritional research. However, some questions remain unanswered. Further research could take a deeper look into the impact of different distributions for the prior of the LME-B within the context of nutritional science under small sample sizes. A recent simulation study by Moeyaert et al. [41] found that adequate priors could improve the performance of LME-B with samples sizes as small as 3 subjects. This would allow LME-B to be used to analyze even very small N-of-1 trials, addressing the shortcoming of needing at least 10 subjects. It remains unclear how this performance will hold under more complex research designs (i.e. time trend and more covariates) and realistic research settings. This deserves further attention to expand our understanding of performance of LME-B, LME-F and IPD and enhance their applicability. An important direction to follow in future research is to combine the points of interest mentioned; what is the number of subjects and/or timepoints needed, and how can we influence this with priors, on model more complex than the one or two parameter models used in prior simulation studies.

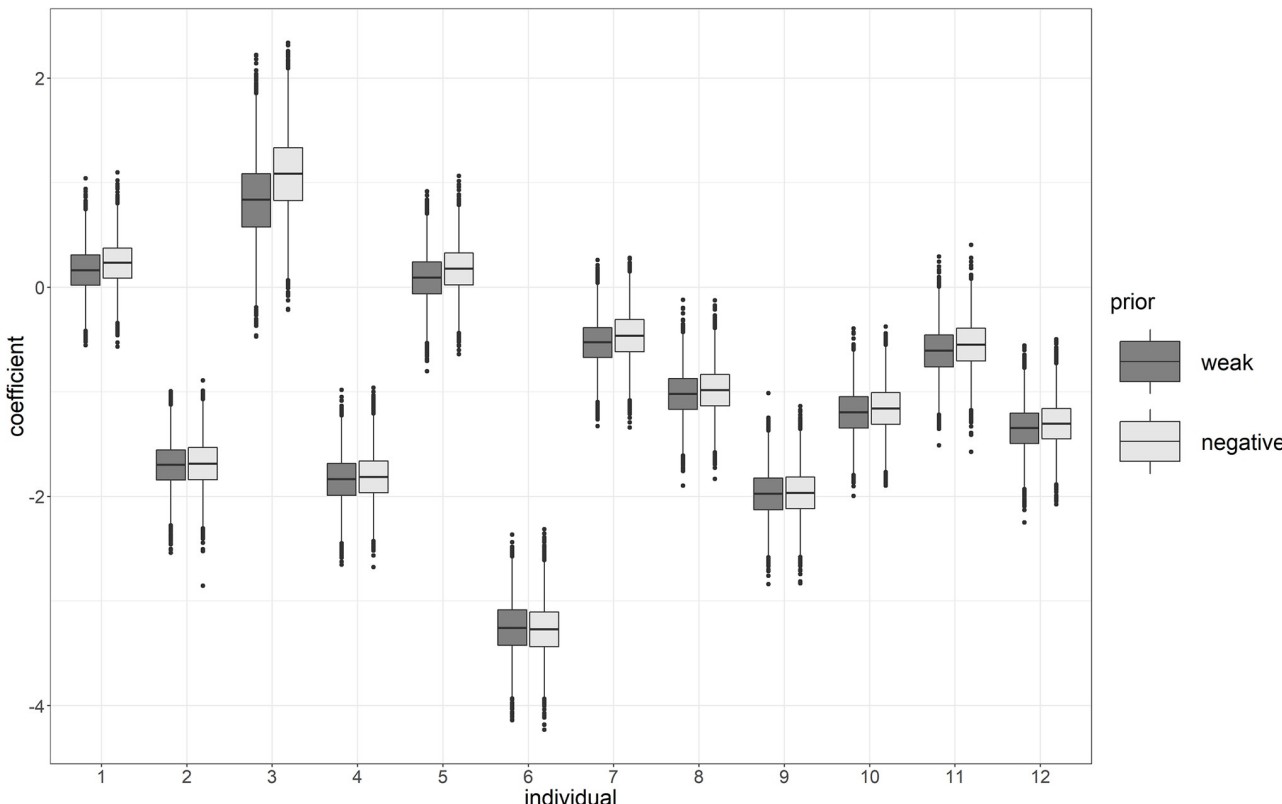

**Fig 9. Individual estimates for the weight loss using a weak and a negative prior, condition 3 versus 1.**

Here, individual differences and the distribution of these individual differences may also play an important role. A formalized framework would thus ease the progression of the individual trials in nutritional and other humane research.

The current paper gives an illustration of the possibilities of N-of-1 designs and which statistical analysis approaches can be used. The N-of-1 studies can be used to investigate the effect of, for instance, a nutritional product on a certain outcome measure at a population level, and, at the same time, to help define a personalized advice. Furthermore, the N-of-1 studies and self-measurement trials bring great new opportunities in the field of food and nutrition, as well as in other fields. To facilitate this progress, research is needed on current and innovative methods and their limitations and implications.

## Supporting information

**S1 Data.**
(CSV)

**S1 Script paper.**
(R)

## Acknowledgments

The authors express their gratitude to Rinke Klein-Entink, Carina Rubingh and Stef van Buuren for their thoughts and contributions to the first version of the manuscript, as well as Albert de Graaf and André Boorsma for the use of their data.

## Author Contributions

**Conceptualization:** Nard D. S. Clabbers, Wilrike J. Pasman.

**Formal analysis:** Tanja Krone, Robin van Stokkum.

**Methodology:** Tanja Krone, Ruud Boessen.

**Project administration:** Sabina Bijlsma.

**Writing – original draft:** Tanja Krone, Ruud Boessen, Sabina Bijlsma, Robin van Stokkum, Nard D. S. Clabbers, Wilrike J. Pasman.

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
