## [Decision Letter · Decision Letter 0]

21 Nov 2019

PONE-D-19-18043

The possibilities of the use of N-of-1 and do-it-yourself trials in nutritional research

PLOS ONE

Dear Mrs Bijlsma,

Thank you for submitting your manuscript to PLOS ONE. After careful consideration, we feel that it has merit but does not fully meet PLOS ONE’s publication criteria as it currently stands. Therefore, we invite you to submit a revised version of the manuscript that addresses the points raised during the review process.

We would appreciate receiving your revised manuscript by Jan 05 2020 11:59PM. To enhance the reproducibility of your results, we recommend that if applicable you deposit your laboratory protocols in protocols.io, where a protocol can be assigned its own identifier (DOI) such that it can be cited independently in the future. For instructions see: http://journals.plos.org/plosone/s/submission-guidelines#loc-laboratory-protocols

We look forward to receiving your revised manuscript.

Kind regards,

Mark Simmonds

Academic Editor

PLOS ONE

Journal Requirements:

1. We note that you have indicated that data from this study are available upon request. PLOS only allows data to be available upon request if there are legal or ethical restrictions on sharing data publicly. For more information on unacceptable data access restrictions, please see http://journals.plos.org/plosone/s/data-availability#loc-unacceptable-data-access-restrictions.

Additional Editor Comments (if provided):

While I agree with the reviewer in general, I am afraid that I find many flaws in your paper, which I set out below.

Broadly, you do not seem to describe or use the methods you propose accurately, which makes me question the validity of your conclusions. In particular, it is unclear what this paper adds to the (currently superior) work of Zucker et al.

While these issues could be resolved by a revision, I stress that that is not certain, and publication after revision cannot be guaranteed.

Major issues:

1) "Meta-analysis" is a broad term that covers any type of statistical combination of studies; so ALL the methods you include are meta-analyses. You seem to be confusing "study-level" (or "aggregate data") meta-analysis with "individual participant data (IPD)" meta-analysis (NOTE the term "individual-subject" is not generally used).

2) IPD meta-analysis generally uses mixed effects modeling, so your terminology is confused. As such I would not recommend study-level meta-analysis of N of 1 trials (except in cases where the participant-level data is unavailable)

2) "Study-level" meta-analysis does not assume all individuals (in N of 1 trials) have the same effect. This is clear if you read DerSimonian and Laird.

3)Zucker et al consider a much more detailed range of models than your paper. This includes various random-effects correlation structures, and a range of Bayesian priors. You must either consider these various models, or make clear why you are using a more restricted choice.

4) Given the above points, you must set out the models your are proposing with much greater detail and clarity than in the current paper.

4) It isn't clear to me how you simulated studies, but I assume a linear mixed effects structure was used. Obviously then, any mixed effect modeling that matches this structure will give the best results. You should be careful that your conclusions are not a consequence of how data were simulated.

Reviewers' comments:

Reviewer's Responses to Questions

**Comments to the Author**

1. Is the manuscript technically sound, and do the data support the conclusions?

Reviewer #1: Yes

2. Has the statistical analysis been performed appropriately and rigorously? 

Reviewer #1: Yes

3. Have the authors made all data underlying the findings in their manuscript fully available?

Reviewer #1: Yes

4. Is the manuscript presented in an intelligible fashion and written in standard English?

Reviewer #1: Yes

5. Review Comments to the Author

Reviewer #1: Thank you for the opportunity to review The Possibilities of the use of N-of-1 and do-it-yourself trials in nutritional research. I enjoyed reading the manuscript, and believe it will make a valuable contribution. However, I do have a few concerns that I will list in the remainder of this review.

1. Although one way of conducting an ISMA is to use the model defined in Equation 1 assuming homogeneous treatment effects, I think other models and assumptions are also possible. Similarly, mixed effect modeling allows for a variety of models, assumptions, and estimation methods. In the discussion (see lines 377 to 385) the authors do a good job of acknowledging that model assumptions may be violated and more complex models could be estimated. It may be helpful to acknowledge some of these complexities in the initial presentation of the three approaches, so that it doesn’t appear that there are three specific models, but rather there are three methodological approaches, each with potential variations.

2. Related to Point 1, I found myself wanting a rationale for why the specific model shown in Equations 1 through 4 was chosen for the simulation. I think it was a reasonable choice, but also think it is helpful to be transparent about when and why choices are made.

3. It would be helpful to formally define the meta-analytic model that was used to synthesize the effect sizes for the ISMA.

4. Also related to Points 1, 2, and 3, I found myself wondering how much of the difference in the simulation results can be attributed to the methodological approach (2 steps where effects were first estimated and then meta-analyzed) versus to the difference in assumptions (e.g., homogeneity of treatment effects versus heterogeneity of treatment effects)?

5. On line 90 should “The most important different” be “The most important difference”?

6. PLOS authors have the option to publish the peer review history of their article (what does this mean?). If published, this will include your full peer review and any attached files.

Reviewer #1: No

---

## [Author Response · Author response to Decision Letter 0]

23 Mar 2020

We uploaded a response to reviewer document. We also uploaded the data we used in the manuscript and a R script to reproduce all results.

---

## [Decision Letter · Decision Letter 1]

21 Apr 2020

The possibilities of the use of N-of-1 and do-it-yourself trials in nutritional research

PONE-D-19-18043R1

Dear Dr. Bijlsma,

We are pleased to inform you that your manuscript has been judged scientifically suitable for publication and will be formally accepted for publication once it complies with all outstanding technical requirements.

With kind regards,

Mark Simmonds

Academic Editor

PLOS ONE

Additional Editor Comments (optional):

Reviewers' comments:

Reviewer's Responses to Questions

**Comments to the Author**

1. If the authors have adequately addressed your comments raised in a previous round of review and you feel that this manuscript is now acceptable for publication, you may indicate that here to bypass the “Comments to the Author” section, enter your conflict of interest statement in the “Confidential to Editor” section, and submit your "Accept" recommendation.

Reviewer #1: All comments have been addressed

2. Is the manuscript technically sound, and do the data support the conclusions?

Reviewer #1: Yes

3. Has the statistical analysis been performed appropriately and rigorously? 

Reviewer #1: Yes

4. Have the authors made all data underlying the findings in their manuscript fully available?

Reviewer #1: Yes

5. Is the manuscript presented in an intelligible fashion and written in standard English?

Reviewer #1: Yes

6. Review Comments to the Author

Reviewer #1: Thank you for your revision of this manuscript. All questions initially posed were addressed, and I think the manuscript will make a helpful contribution to the literature.

7. PLOS authors have the option to publish the peer review history of their article (what does this mean?). If published, this will include your full peer review and any attached files.

Reviewer #1: No

---

## [Editor Report · Acceptance letter]

24 Apr 2020

PONE-D-19-18043R1 

The possibilities of the use of N-of-1 and do-it-yourself trials in nutritional research 

Dear Dr. Bijlsma:

I am pleased to inform you that your manuscript has been deemed suitable for publication in PLOS ONE. Congratulations! Your manuscript is now with our production department. 

With kind regards,

on behalf of

Dr. Mark Simmonds 

Academic Editor

PLOS ONE